# Increased Complement Activation in Systemic Sclerosis Patients with Skin and Lung Fibrosis

**DOI:** 10.3390/jpm12020284

**Published:** 2022-02-15

**Authors:** Chiara Pellicano, Marzia Miglionico, Laura Romaggioli, Amalia Colalillo, Lorenzo Vantaggio, Cecilia Napodano, Cinzia Callà, Francesca Gulli, Mariapaola Marino, Umberto Basile, Edoardo Rosato

**Affiliations:** 1Department of Translational and Precision Medicine, Sapienza University of Rome, 00185 Rome, Italy; chiara.pellicano@gmail.com (C.P.); mrz.miglionico@gmail.com (M.M.); laura.romaggioli@gmail.com (L.R.); amalia.colalillo@uniroma1.it (A.C.); lorenzo.vantaggio@uniroma1.it (L.V.); edoardo.rosato@uniroma1.it (E.R.); 2Synlab Data Medica, 35133 Padova, Italy; cecilia.napodano@gmail.com; 3Dipartimento di Scienze di Laboratorio e Infettivologiche, Fondazione Policlinico Universitario “A. Gemelli” IRCCS, Università Cattolica del Sacro Cuore, 00168 Rome, Italy; cinziaanna.calla@unicatt.it (C.C.); umberto.basile@policlinicogemelli.it (U.B.); 4Laboratorio di Patologia Clinica, Ospedale Madre Giuseppina Vannini, 00177 Rome, Italy; dottfgulli@gmail.com; 5Dipartimento di Medicina e Chirurgia Traslazionale, Sezione di Patologia Generale, Fondazione Policlinico Universitario “A. Gemelli” IRCCS, Università Cattolica del Sacro Cuore, 00168 Rome, Italy

**Keywords:** systemic sclerosis, complement, interstitial lung disease, digital ulcers

## Abstract

Introduction: The involvement of complement system in the phenotypic expression of systemic sclerosis (SSc) is a debated topic. We aimed to assay complement fractions in SSc patients and to correlate their levels with the clinical course of disease. Key points: 1. CH50 is increased in SSc patients compared to HC; 2. Serum C2 levels are increased in SSc patients compared to HC; 3. CH50 may represent a biomarker of skin and lung fibrosis severity in SSc patients. Method: Complement hemolysis 50% (CH50), C2, C3 and C4 levels have been assessed in 85 SSc patients and 47 healthy controls (HC). Results: SSc patients displayed a statistically significant higher value of CH50 [76.3 U/mL (IQR 65.8–89.4 U/mL) vs. 29.6 U/mL (IQR 24.7–34 U/mL); *p* < 0.0001] and of C2 [26.1 mg/L (IQR 24.1–32.1 mg/L) vs. 22.7 mg/L (IQR 20.6–24.4 mg/L); *p* < 0.0001] if compared to HC. Patients with diffuse cutaneous SSc (dcSSc) had higher levels of CH50 than patients with limited cutaneous SSc (lcSSc) [83.6 U/mL (IQR 72.3–102.7 U/mL) vs. 71.3 U/mL (IQR 63.7–83.6 U/mL); *p* = 0.003]. SSc patients with interstitial lung disease (ILD) had higher CH50 levels if compared to SSc patients without ILD [79.6 U/mL (IQR 68.3–97.4 U/mL) vs. 69.7 U/mL (54.6–85.7 U/mL); *p* = 0.042]. A positive linear correlation existed between CH50 and the modified Rodnan Skin Score (mRSS) (r = 0.285, *p* = 0.008) and disease severity scale (DSS) (r = 0.285, *p* = 0.005); a negative linear correlation was demonstrated between CH50 and the diffusing capacity of carbon monoxide (DLco) (r = −0.252, *p* = 0.012). In multiple linear regression analysis, only DSS was significant (*p* = 0.01, beta coefficient 2.446). Conclusions: Our results show an increment of CH50 and serum C2 levels in SSc patients in comparison to HC; we retain that CH50 may represent a biomarker of disease severity and of skin and lung fibrosis in these patients.

## 1. Introduction

Systemic sclerosis (SSc) is a systemic autoimmune disease that leads to skin fibrosis and internal organs. Visceral involvement is heterogeneous and results in significant morbidity and mortality [1]. The pathogenesis of SSc is still unclear, but vascular injury and immune system dysregulation are typical features of the disease. Microvascular dysfunction with endothelial cells (ECs) damage is responsible for the activation of B-lymphocytes, autoantibodies production, and hyperactivity of T cells [2]. At the same time, the activation of immune system leads to the release of proinflammatory cytokines that provoke vascular injury together with fibroblast activation and proliferation with collagen deposition [3,4,5,6]. During the reperfusion phase of ischemic/reperfusion injury, which characterizes Raynaud’s phenomenon, complement activation occurs with the attack of ECs as non-self-antigens, worsening the endothelial damage [7,8]. The complement system, composed of a complex cascade of circulating and surface-bound proteins, is able to activate coagulation cascade and angiogenesis, and it seems to be essential for the integrity of ECs [9]. An over- or poorly controlled complement activation causes altered opsonization and recruitment of inflammatory cells, with cell lysis and immune complex clearance. This cascade provokes EC damage and apoptosis, but also increases the expression of vascular cell adhesion molecules, enhancing the local immune response [10]. Alterations of the complement system have also been demonstrated in rheumatoid arthritis (RA) and systemic lupus erythematosus (SLE) [11,12]. In many patients with active and severe SLE, low levels of circulating C3 and C4 components are detectable, indicating the consumption of complement cascade via the classical pathway through immune complex formation [13]. Moreover, altered Toll-like receptor (TLR) signal pathways, described in SSc patients, could be the trigger for the activation of complement cascade leading to hypocomplementemia, and may play a role in the pathogenesis of disease. However, it is still unclear whether complement activation and consumption may correlate with disease activity [13].

The β1-glycoprotein C2 is a component of the complement system that plays a key role in the classical and mannose-binding lectin pathways of activation. Activated C1s are responsible for the cleavage of C2 into two fragments, C2a and C2b. C2a, the larger one, combines with C4b to produce C3 or C5 convertase, that are essential for the activation of the terminal pathway of complement cascade [14].

The complement hemolysis 50% (CH50) is a screening assay that provides a functional measurement of activation of the classical complement pathway, the immunoglobulins-mediated one, activated in the course of several inflammatory diseases. During chronic or acute inflammation, CH50 levels are usually increased [14].

The primary aim of this study is to assess CH50 and the serum level of C2, C3 and C4 in patients affected by SSc and in healthy controls (HC). The secondary aim is to evaluate the role of complement cascade in clinical features of disease.

## 2. Materials and Methods

### 2.1. Subjects

We enrolled 85 SSc patients [F = 76, median age 56 years (IQR 48–64 years)], responding to the American College of Rheumatology/European League Against Rheumatism Collaborative Criteria for SSc [15]. According to Le Roy et al., 34 (40%) had diffuse cutaneous SSc (dcSSc), and 51 (60%) limited cutaneous SSc (lcSSc) [15]. All patients underwent vasodilator therapy with endoprost, and none of them underwent immunosuppressive therapy or high-dose corticosteroids (equivalent to prednisone > 10 mg) in the past 6 months before enrollment. All patients continued stable therapy from enrollment until clinical evaluation and serum complement assay. Corticosteroids were used by 30 SSc patients (5 mg of prednisone daily). Table 1 summarizes the demographic and clinical features of SSc patients. Forty-seven HC matched for sex and age were also enrolled in this study.

Autoimmune overlap syndromes, glomerulonephritis, solid or haematological cancer, hepatic, renal or cardiac failure not related to SSc, immunosuppressive therapy or high-dose corticosteroids (equivalent to prednisone > 10 mg) in the past 6 months, smoke, pregnancy and breastfeeding, were considered as exclusion criteria.

The subjects’ written consent was obtained, and the study was conducted according to the Declaration of Helsinki. The study was approved by the ethics committee of Sapienza University of Rome (IRB n 0304).

### 2.2. Clinical Correlates of SSc Patients

The modified Rodnan skin score (mRSS) has been used to assess skin involvement and the disease subset (dcSSc or lcSSc) [16,17], and digital ulcers (DUs) were defined according Amanzi et al. [18]

The disease activity index (DAI) was calculated to assess disease activity, following indications of the European Scleroderma Trials and Research group (EUSTAR) task force for the development of revised EUSTAR criteria for systemic sclerosis [19]. This index consists of six variables with different weight: change in skin sclerosis in the last month, presence of DUs, mRSS > 18, presence of tendon friction rubs, diffusing capacity of the lung for monoxide carbon (DLco) < 70% of the predicted value, and C-Reactive Protein (CRP) > 1 mg/dl. 

Severity of disease was evaluated through the disease severity scale (DSS) [20]. This index is based upon the evaluation of nine elements: general state, peripheral vessels, skin, joints/tendons, muscles, gastrointestinal tract, lungs, heart, and kidneys. Each organ or system is assessed separately with a score ranging from a minimum of 0 to a maximum of 4, in consideration of its null, intermediate, moderate, severe or “end-stage” involvement.

Nailfold videocapillaroscopy (NVC) was performed at the level of the distal phalanx of the second, third and fourth fingers of both hands using a videocapillaroscope equipped with a 500* magnification lens (Pinnacle Studio Version 8 software, Corel, Ottawa, Canada). According to Cutolo et al., the capillaroscopic images were classified according to the patterns of early, active, and late [16].

The antibody profile was evaluated by indirect immunofluorescence (IIF) for the detection of antinuclear antibodies (ANA) and definition of the IF pattern (homogeneous, speckled, centromeric, cytoplasmic), and ELISA for the detection of SSc-specific antibodies (Scl70). 

A transthoracic echocardiocolordoppler with a General Electric Vivid S5 apparatus (GE Medical Systems, Haifa, Israel Ltd.) was performed for the evaluation of the main echocardiographic parameters (diameter of the left ventricle, septum thickness, ejection fraction, diameter of the right ventricle, systolic excursion of the tricuspid, area of the right and left atrium) and estimation of systolic pulmonary arterial pressure (sPAP) [21]. 

Respiratory function tests were performed to evaluate the forced expiratory volume in the first minute (FEV1), forced vitality capacity (FVC) and DLco. All spirometric parameters were recorded with a Quark PFT 2 (Cosmed) spirometer and expressed according to the standards recommended by the American/European Respiratory Society [22].

We also considered high-resolution computed tomogrpy (HRCT) or, if carried out within the previous 3 months, image collection. The images were classified in radiological patterns: normal, ground glass, reticular, and honeycombing. Interstitial lung disease (ILD) was defined according to Goh et al. [23].

### 2.3. Evaluation of Complement Cascade

Complement C3, C4, C2 fractions and CH50 were evaluated by turbidimetry. Peripheral venous blood samples were collected in tubes containing sodium citrate or EDTA and centrifuged at 3000× *g* for 15 min at 19 °C. Serum samples were aliquoted into 1.5 mL Eppendorf tubes and stored at −80 °C until the time of assay. The determination of C3 and C4 levels by turbidimetric methods involves the reaction with specific antiserum to form insoluble complexes. When light crosses the suspension, a portion of the light is transmitted and focused onto a photodiode by an optical lens system. The amount of transmitted light is indirectly proportional to the specific protein amounts in the test sample. Concentrations are automatically calculated by reference to a calibration curve stored within the instrumentt Optilite turbidimeter (The Binding Site Ltd., Birmingham, UK). The automated Human Complement C2 assay was used to measure C2 levels, and a CH50 assay was performed to evaluate CH50 classical pathway activity. Both assays consist in automated turbidimetric assays and were run on the Optilite turbidimeter (The Binding Site Ltd., Birmingham, UK). The traditional method used to measure total complement activity in serum is the CH50 test, based on complement-mediated capability to lyse sheep red blood cells pre-coated with rabbit anti-sheep red blood cell antibody (haemolysin). The Optilite CH50 assay ameliorates these steps by directly testing the function of the membrane attack complex, thereby quantifying total complement activity. Samples were tested according to the manufacturer’s instructions, and serum dilutions were performed according to the manufacturer’s recommendations when necessary. An operator blinded to clinical information performed the assays. For the repeatability of the manufacturer’s methods, we followed the guidelines of the Clinical and Laboratory Standard Institute [24].

### 2.4. Statystical Analysis

SPSS version 25.0 software (Bioz, Los Altos, CA) was used for statistical analysis. After an evaluation of normality by using the Shapiro–Wilk test, continuous variables were expressed as median and IQR. Differences between groups were evaluated by Student’s or Mann-Whitney’s *t*-test. Differences between categorical variables were evaluated by the chi-square or Fisher’s exact test. The Pearson or Spearman correlation test was used for bivariate correlations. Stepwise multiple regression analysis was used to evaluate the correlation between CH50 and C2 with disease variables (mRSS, DAI, DSS, ESR, CRP, sPAP, DLco). A *p*-value < 0.05 was considered significant.

## 3. Results

### 3.1. Evaluation of CH50 and Complement Fractions in SSc Patients 

SSc patients displayed a statistically significant higher median value of CH50 than HC [76.3 U/mL (IQR 65.8–89.4 U/mL) vs. 29.6 U/mL (IQR 24.7–34 U/mL); *p* < 0.0001] (Figure 1A). Moreover, the median value of C2 was significantly higher in SSc patients compared to HC [26.1 mg/L (IQR 24.1–32.1 mg/L) vs. 22.7 mg/L (IQR 20.6–24.4 mg/L); *p* < 0.0001]. There were no differences in C3 and C4 levels between SSc patients and HC (1.07 g/L (IQR 0.99–1.12 g/L) vs. 0.84 g/L (IQR 0.66–0.98 g/L) and 0.2 g/L (IQR 0.16–0.23 g/L) vs. 0.36 g/L (IQR 0.27–0.41 g/L), respectively; *p* > 0.05). We report the entire results in Table 2.

### 3.2. Analysis of Disease Variables and Complement Assessment in SSc Patients

Age and disease duration did not correlate with CH50, C2, C3, C4, ESR and CRP (*p* > 0.05).

We did not find any differences between male and female SSc patients in CH50 [79.82 U/mL (IQR 63.79–82.26 U/mL) vs. 75.9 U/mL (IQR 66–90.57 U/mL); *p* > 0.05], C2 [24.4 mg/L (IQR 23.1–27.3 mg/L) vs. 26.3 mg/L (IQR 24.2–32.7 mg/L); *p* > 0.05], C3 [1.18 g/L (IQR 1.07–1.27 g/L) vs. 1.13 g/L (IQR 1–1.23 g/L); *p* > 0.05], C4 [0.22 g/L (IQR 0.2–0.23 g/L) vs. 0.2 g/L (IQR 0.16–0.23 g/L); *p* > 0.05], ESR [16 mm/h (IQR 8–26 mm/h) vs. 22 mm/h (IQR 12–31 mm/h); *p* > 0.05] and CRP [2000 mcg/L (IQR 900–3700 mcg/L) vs. 1800 mcg/L (IQR 1000–4600 mcg/L); *p* > 0.05].

SSc patients with dcSSc had statistically higher levels of CH50 than SSc patients with lcSSc [83.6 U/mL (IQR 72.3–102.7 U/mL) vs. 71.3 U/mL (IQR 63.7–83.6 U/mL); *p* = 0.003]. No differences emerged in C2 levels according to disease subset [25.8 mg/L (IQR 24.2–31 mg/L) vs. 27 mg/L (IQR 23.7–32.3 mg/L); *p* > 0.05]. Moreover, no statistically significant differences were found in C3 [1.15 g/L (IQR 1.07–1.29 g/L) vs. 1.1 g/L (IQR 0.99–1.21 g/L); *p* > 0.05], C4 [0.2 g/L (IQR 0.18–0.24 g/L) vs. 0.2 g/L (IQR 0.16–0.23 g/L); *p* > 0.05], ESR [25 mm/h (IQR 16–33 mm/h) vs. 17 mm/h (IQR 11–29 mm/h); *p* > 0.05] and CRP [2500 mcg/L (IQR 950–6550 mcg/L) vs. 1600 mcg/L (IQR 1000–3100 mcg/L); *p* > 0.05] between dcSSc patients and lcSSc patients.

There was no statistically significant difference according to NVC patterns (early, active and late) in CH50 [66.21 U/mL (IQR 58.18–80.49 U/mL) vs. 73.72 U/mL (IQR 63.79–87.68 U/mL) vs. 79.82 U/mL (IQR 69.74–101.6 U/mL); *p* > 0.05], C2 [25.8 mg/L (IQR 23.7–27.3 mg/L) vs. 29.5 mg/L (24.9–35.4 mg/L) vs. 25.7 mg/L (23.9–30.8 mg/L); *p* > 0.05], C3 [1.1 g/L (IQR 1.07–1.24 g/L) vs. 1.07 g/L (IQR 0.97–1.2 g/L) vs. 1.14 g/L (IQR 1.05–1.23 g/L); *p* > 0.05], C4 [0.21 g/L (IQR 0.18–0.23 g/L) vs. 0.2 g/L (IQR 0.1–0.29 g/L) vs. 0.26 g/L (IQR 0.16–0.37 g/L); *p* > 0.05], ESR [12 mm/h (IQR 8–19 mm/h) vs. 20 mm/h (IQR 10–29 mm/h) vs. 26 mm/h (IQR 16–37 mm/h); *p* > 0.05] and CRP [1600 mcg/L (IQR 900–2300 mcg/L) vs. 1400 mcg/L (IQR 700–3200 mcg/L) vs. 2500 mcg/L (IQR 1100–5900 mcg/L); *p* > 0.05].

Moreover, no statistically significant difference was found according to autoantibodies specificity (Scl70, centromere and ANA) in CH50 [80.59 U/mL (IQR 71.52–101.6 U/mL) vs. 71.77 U/mL (IQR 65.31–85.68 U/mL) vs. 73.88 U/mL (IQR 58.14–87.68 U/mL); *p* > 0.05], C2 [25.8 mg/L (IQR 24.7–31 mg/L) vs. 27.8 mg/L (25–32.3 mg/L) vs. 26 mg/L (23.1–32.1 mg/L); *p* > 0.05], C3 [1.15 g/L (IQR 1.1–1.29 g/L) vs. 1.1 g/L (IQR 1.01–1.2 g/L) vs. 1.13 g/L (IQR 0.99–1.21 g/L); *p* > 0.05], C4 [0.21 g/L (IQR 0.18–0.23 g/L) vs. 0.2 g/L (IQR 0.16–0.25 g/L) vs. 0.2 g/L (IQR 0.15–0.23 g/L); *p* > 0.05], ESR [22 mm/h (IQR 11–29 mm/h) vs. 18 mm/h (IQR 10–29 mm/h) vs. 22 mm/h (IQR 13–36 mm/h); *p* > 0.05] and CRP [2150 mcg/L (IQR 850–5650 mcg/L) vs. 1400 mcg/L (IQR 700–2000 mcg/L) vs. 2950 mcg/L (IQR 1050–5550 mcg/L); *p* > 0.05].

SSc patients with ILD had higher CH50 levels compared to SSc patients without ILD [79.6 U/mL (IQR 68.3–97.4 U/mL) vs. 69.7 U/mL (54.6–85.7 U/mL); *p* = 0.042]. C2 serum levels were similar between SSc patients with ILD and SSc patients without ILD [25.8 mg/L (IQR 24–32.5 mg/L) vs. 28.3 mg/L (24.1–32.1 mg/L); *p* > 0.05]. Moreover, no statistically significant differences were found in C3 [1.14 g/L (IQR 1.01–1.24 g/L) vs. 1.12 g/L (IQR 1–1.2 g/L); *p* > 0.05], C4 [0.2 g/L (IQR 0.16–0.23 g/L) vs. 0.2 g/L (IQR 0.15–0.27 g/L); *p* > 0.05], ESR [23 mm/h (IQR 13–37 mm/h) vs. 16 mm/h (IQR 10–26 mm/h); *p* > 0.05] and CRP [1900 mcg/L (IQR 1000–5200 mcg/L) vs. 1750 mcg/L (IQR 700–2600 mcg/L); *p* > 0.05] between SSc patients with ILD and SSc patients without ILD.

SSc patients with a DU history had higher CH50 levels compared to SSc patients without a DU history [80.4 U/mL (IQR 68.3–101.07 U/mL) vs. 71.39 U/mL (62.91–84 U/mL); *p* = 0.014]. No differences emerged in C2 [25.8 mg/L (IQR 24–32.8 mg/L) vs. 27.6 mg/L (24.1–32.1 mg/L); *p* > 0.05], C3 [1.14 g/L (IQR 1.05–1.23 g/L) vs. 1.1 g/L (IQR 0.97–1.24 g/L); *p* > 0.05], C4 [0.2 g/L (IQR 0.18–0.23 g/L) vs. 0.2 g/L (IQR 0.16–0.23 g/L); *p* > 0.05], ESR [22 mm/h (IQR 12–39 mm/h) vs. 19 mm/h (IQR 11–29 mm/h); *p* > 0.05] and CRP [2400 mcg/L (IQR 1000–5200 mcg/L) vs. 1600 mcg/L (IQR 900–2600 mcg/L); *p* > 0.05] between SSc patients with a DU history and SSc patients without a DU history.

SSc patients with active DUs had similar CH50 [67.19 U/mL (IQR 49.72–76.32 U/mL) vs. 78.24 U/mL (IQR 66.21–91.75 U/mL); *p* > 0.05], C2 [25.8 mg/L (IQR 24–32.8 mg/L) vs. 27.6 mg/L (24.1–32.1 mg/L); *p* > 0.05], C3 [1.14 g/L (IQR 1.05–1.23 g/L) vs. 1.1 g/L (IQR 0.97–1.24 g/L); *p* > 0.05], C4 [0.2 g/L (IQR 0.18–0.23 g/L) vs. 0.2 g/L (IQR 0.16–0.23 g/L); *p* > 0.05], ESR [22 mm/h (IQR 12–39 mm/h) vs. 19 mm/h (IQR 11–29 mm/h); *p* > 0.05] and CRP [2400 mcg/L (IQR 1000–5200 mcg/L) vs. 1600 mcg/L (IQR 900–2600 mcg/L); *p* > 0.05] than SSc patients without active DUs.

Twelve SSc patients showed joint/tendon involvement. No statistically significant differences were found in C3 [1.19 g/L (IQR 1.11–1.27 g/L) vs. 1.12 g/L (IQR 1.01–1.23 g/L); *p* > 0.05], C4 [0.2 g/L (IQR 0.16–0.26 g/L) vs. 0.2 g/L (IQR 0.16–0.23 g/L); *p* > 0.05], C2 [25 mg/L (IQR 22.6–25.7 mg/L) vs. 27.2 mg/L (24.2–32.5 mg/L); *p* > 0.05] and CH50 [83.86 U/mL (IQR 74.6–103.18 U/mL) vs. 74.05 U/mL (IQR 65.31–86.86 U/mL); *p* > 0.05] between SSc patients with or without joint/tendon involvement.

Seven SSc patients had pulmonary arterial hypertension (PAH). SSc patients with PAH had similar CH50 [79.24 U/mL (IQR 74.05–110.85 U/mL) vs. 75.86 U/mL (IQR 65.46–87.68 U/mL); *p* > 0.05], C2 [25 mg/L (IQR 22.4–25.7 mg/L) vs. 27 mg/L (24.1–32.3 mg/L); *p* > 0.05], C3 [1.13 g/L (IQR 0.85–1.26 g/L) vs. 1.13 g/L (IQR 1.01–1.23 g/L); *p* > 0.05], C4 [0.19 g/L (IQR 0.16–0.28 g/L) vs. 0.2 g/L (IQR 0.16–0.23 g/L); *p* > 0.05] than SSc patients without PAH.

No statistically significant differences were found in C3 [1.14 g/L (IQR 1.03–1.24 g/L) vs. 1.12 g/L (IQR 0.99–1.23 g/L); *p* > 0.05], C4 [0.2 g/L (IQR 0.16–0.25 g/L) vs. 0.2 g/L (IQR 0.16–0.23 g/L); *p* > 0.05], C2 [26.4 mg/L (IQR 24.7–32.5 mg/L) vs. 26.1 mg/L (23.3–32.1 mg/L); *p* > 0.05] and CH50 [75.19 U/mL (IQR 64.82–98.47 U/mL) vs. 79.24 U/mL (IQR 66.21–87.68 U/mL); *p* > 0.05] between SSc patients undergoing steroid therapy and SSc patients not treated with steroids.

A positive linear correlation was shown between CH50 and mRSS (r = 0.285, *p* = 0.008) and CH50 and DSS (r = 0.285, *p* = 0.005) (Figure 1B,C); we also found a positive linear correlation between CH50 and DAI (r = 0.261, *p* = 0.009), CH50 and CRP (r = 0.260, *p* = 0.01), CH50 and ESR (r = 0.249, *p* = 0.012) and CH50 and sPA) (r = 0.227, *p* = 0.021). A negative linear correlation was demonstrated between CH50 and DLco (r = −0.252, *p* = 0.012) (Figure 1D). No correlation was found between CH50 levels and serum immunoglobulins (IgG, IgA and IgM) and FVC (*p* > 0.05). In bivariate analysis, C2, C3 and C4 levels showed no correlation with any clinical variable (mRSS, DSS, DAI, sPAP, DLco, FVC) (*p* > 0.05). All the correlations in bivariate analysis are summarized in Table 3.

In the stepwise multiple linear regression analysis, all the significant variables in the bivariate analysis were included; the only variable that remained significant was DSS (*p* = 0.01, beta coefficient 2.446).

Results are displayed in Table 4.

## 4. Discussion

The role of complement in the clinical manifestations of SSc has been poorly analyzed. A limited analysis of C3 and C4 complement fractions was carried out in most cases [13]. Hypocomplementemia (C3 and C4) was an item of the first EUSTAR group activity index, used to assess the activity of diseases with a weight of 1.0 in the 10-point indices of disease activity preliminarily evaluated [25]. This index would define active to very active disease with sensitivity ranging from 62 to 81% and specificity ranging from 86 to 93% [25]. However, the role of hypocomplementemia in assessing SSc activity was still controversial [26]. The revised EUSTAR activity index differs from the original for the exclusion of hypocomplementemia: it was not associated with disease activity in the study, even in univariate analysis [19].

Senaldi et al. demonstrated that complement activation occurs in SSc patients, with the highest level of activation being characteristic of dcSSc: serum levels of C3d, C3d:C3, Ba, and Ba:factor B were higher in dcSSc patients compared to lcSSc patients [27].

To our knowledge, only a few reports have described increased levels of C2 fraction and of the last fractions of the complement cascade (C5–C9) in SSc patients compared to HC [28]. Interestingly, Venneker et al. studied the expression of the membrane cofactor protein (MCP) and decay-accelerating factor (DAF) in the endothelium of skin biopsies of SSc patients, both in lesional and non-lesional samples, and in the endothelium of morphea lesions. Moreover, they compared these samples with samples of HC and patients with other autoimmune diseases. MCP and DAF expression were altered only in SSc patients, suggesting defective endothelial protection, due to a reduced expression of complement regulatory proteins [29,30]. These results were confirmed by Scambi et al. which detected the abnormal deposition of the membrane attack complex (MAC) on the endothelium in skin biopsies of SSc patients and reduced MCP expression on the vascular endothelial surface of patients. The authors proposed that the local activation of the complement system on vascular endothelium in association with a lack of the local mechanism of regulation might promote complement activation, leading to MAC subendothelial deposition and EC apoptosis [31].

Here, we analyzed the total complement activity (CH50) in patients affected by SSc, showing that serum C2 levels and CH50 activity were increased in comparison to HC. Our results demonstrated that only CH50 is a potential marker of skin and lung fibrosis, and conversely, C2, C3, and C4 did not show any association with disease variables. In addition, CH50 was increased in SSc patients with a DU history. For the first time, we here evaluated the role of CH50 as an expression of the total complement activity, starting from the assumption that the fractions C3 and C4 were normal. We can hypothesize that the complement cascade is activated in SSc and that the increase in total complement activity is linked to an activation of the last fractions of the complement and not of the classical route, since we did not find consumption of serum levels of C3 and C4 that were normal.

Our hypothesis is that the total complement activity, assessed by CH50, might be associated with skin fibrosis, as highlighted by the fact that complement activation is more evident in the dcSSc patients compared to lcSSc patients, and that CH50 activity positively correlates with mRSS. Although the correlation indices are low, probably due to the small sample analyzed, we can suppose that the activation of the complement is linked to the fibrotic phase following the inflammation. Moreover, larger cohort studies are needed to evaluate the role of complement in skin fibrosis.

From our knowledge, no reports in literature describe the role of complement in the ILD. In this study, we showed that total complement activity is associated with ILD, assessed by both DLco and HRCT, suggesting a pathogenetic role in the ILD. However, further analyses also taking histological features into account are needed to evaluate the role of complement in ILD. 

Moreover, we found that CH50 activity was increased in patients with a DU history and not in patients with active DUs. This could be explained with both the deposition of the complement in the vascular bed, but also with the fact that a DU history is associated with greater vascular damage and loss of capillaries with fibrotic replacement.

We are conscious that there are limitations to our study. The first is the lack of histological findings to evaluate the role of activation of complement in the skin and lung fibrosis. Second, we could not use a computational platform for texture analysis of ILD patterns to quantify parenchymal lung abnormalities.

Our goal, however, was to conduct a pilot study in SSc patients showing that CH50 and serum C2 levels are increased in comparison to HC, and that CH50 may represent a biomarker of disease severity and of skin and lung fibrosis. Further studies are needed to evaluate the role of these preliminary data and to prospectively assess the predictive value of complement activity in disease progression. Proposed biomarkers and their employment in new integrated specific panels for autoimmune diseases represent a challenge for clinical and laboratory investigations.

## Figures and Tables

**Figure 1 jpm-12-00284-f001:**
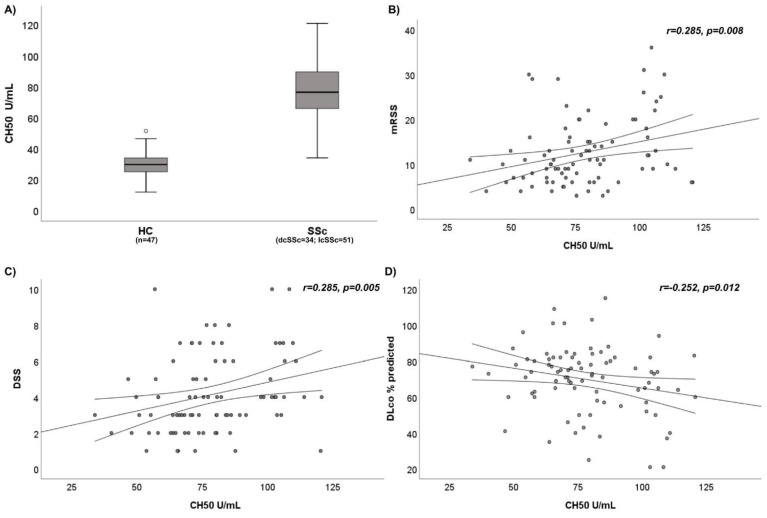
(**A**) Comparison of CH50 activity between SSc patients and HC. (**B**) Correlation between CH50 activity and modified Rodnan Skin Score (mRSS) in SSc patients. (**C**) Correlation between CH50 activity and Disease Severity Scale (DSS) in SSc patients. (**D**) Correlation between CH50 activity and Diffusing Capacity of the lung for monoxide carbon (DLco) in SSc patients.

**Table 1 jpm-12-00284-t001:** Demographic and clinical features of SSc patients.

Age, years, median and IQR	56 (48–64)
Female, *n* (%)	76 (89.4)
dcSSc, *n* (%)	34 (40)
Disease duration, years, median and IQR	12 (7–16)
mRSS, median and IQR	11 (7–16)
SSc-specific autoantibodies:	
Anti-topoisomerase I, *n* (%)	29 (34.1)
Anti-centromere, *n* (%)	22 (25.9)
None, *n* (%)	34 (60)
Nailfold capillaroscopic pattern	
Early, *n* (%)	15 (17.6)
Active, *n* (%)	27 (31.8)
Late, *n* (%)	43 (50.6)
DAI, median and IQR	1.5 (0.8–2.8)
DSS, median and IQR	4 (3–6)
sPAP, mmHg, median and IQR	27 (25–31)
DLco, % of predicted, median and IQR	73 (60–81)
ILD, *n* (%)	62 (72.9)
Active DUs, *n* (%)	7 (8.2)
DUs history, *n* (%)	47 (55.3)
ESR, mm/h, median and IQR	22 (12–29)
CRP, mcg/L, median and IQR	1850 (1000–4600)

SSc: Systemic Sclerosis; dcSSc: diffuse cutaneous Systemic Sclerosis; mRSS: modified Rodnan Skin Score; DAI: Disease Activity Index; DSS: Disease Severity Scale; sPAP: systolic Pulmonary Arterial Pressure; DLco: Diffusing Capacity of the lung for monoxide carbon; ILD: Interstitial Lung Disease; DUs: Digital Ulcers; ESR: Erythrocyte Sedimentation Rate; CRP: C-Reactive Protein; IQR: Interquartile Range.

**Table 2 jpm-12-00284-t002:** Complement fractions and immunoglobulins in SSc patients and HC.

	SSc	HC	*p*
CH50, U/mL, median and IQR	76.3 (65.8–89.4)	29.6 (24.7–34)	*p* < 0.0001 *
C2, mg/L, median and IQR	26.1 (24.1–32.1)	22.7 (20.6–24.4)	*p* < 0.0001 *
C3, g/L, median and IQR	1.07 (0.99–1.12)	0.84 (0.66–0.98)	*p* > 0.05
C4, g/L, median and IQR	0.2 (0.16–0.23)	0.36 (0.27–0.41)	*p* > 0.05
IgG, g/L, median and IQR	10.4 (9.4–12.3)	10.6 (9.2–14.2)	*p* > 0.05
IgA, g/L, median and IQR	2.37 (1.87–3.21)	2.52 (1.93–3.44)	*p* > 0.05
IgM, g/L, median and IQR	1.27 (0.85–1.78)	1.02 (0.78–1.88)	*p* > 0.05

SSc: Systemic Sclerosis; HC: Healthy Controls; IQR: Interquartile Range. *: statistically significant.

**Table 3 jpm-12-00284-t003:** Linear correlation (correlation coefficients = r) between complement fractions and demographic and disease variables.

	Age	Disease Duration	mRSS	DAI	DSS	sPAP	DLco	FVC	ESR	CRP
CH50	r = 0.062, *p* = 0.286	r = 0.025, *p* = 0.411	r = 0.285, *p* = 0.008 *	r = 0.261, *p* = 0.009 *	r = 0.285, *p* = 0.005 *	r = 0.227, *p* = 0.021 *	r = −0.252, *p* = 0.012 *	r = −0.113, *p* = 0.151	r = 0.249, *p* = 0.012 *	r = 0.260, *p* = 0.01 *
C2	r = 0.130, *p* = 0.118	r = 0.099, *p* = 0.184	r = −0.079, *p* = 0.237	r = −0.160, *p* = 0.072	r = −0.116, *p* = 0.146	r = −0.053, *p* = 0.317	r = 0.139, *p* = 0.102	r = 0.042, *p* = 0.350	r = 0.001, *p* = 0.496	r = −0.096, *p* = 0.196
C3	r = −0.060, *p* = 0.293	r = 0.035, *p* = 0.374	r = 0.080, *p* = 0.232	r = 0.107, *p* = 0.165	r = 0.137, *p* = 0.106	r = 0.078, *p* = 0.240	r = −0.129, *p* = 0.119	r = −0.128, *p* = 0.122	r = 0.267, *p* = 0.007 *	r = 0.149, *p* = 0.090
C4	r = 0.058, *p* = 0.300	r = −0.160, *p* = 0.073	r = 0.179, *p* = 0.052	r = 0.057, *p* = 0.304	r = 0.071, *p* = 0.261	r = 0.011, *p* = 0.459	r = −0.090, *p* = 0.208	r = −0.013, *p* = 0.455	r = 0.041, *p* = 0.355	r = 0.059, *p* = 0.299

mRSS: modified Rodnan Skin Score; DAI: Disease Activity Index; DSS: Disease Severity Scale; sPAP: systolic Pulmonary Arterial Pressure; DLco: Diffusing Capacity of the lung for monoxide carbon; FVC: forced vital capacity; ESR: Erythrocyte Sedimentation Rate; CRP: C-Reactive Protein. *: statistically significant.

**Table 4 jpm-12-00284-t004:** Multiple linear regression between dependent variable (CH50) and independent variables.

Variables	Beta Coefficient	Standard Error	*p*
mRSS	0.162	1.189	*p* = 0.238
DAI	0.101	0.589	*p* = 0.558
DSS	2.446	0.926	*p* = 0.01 *
ESR	0.170	1.491	*p* = 0.140
CRP	0.199	1.809	*p* = 0.074
sPAP	0.136	0.861	*p* = 0.228
DLco	−0.117	0.645	*p* = 0.346

mRSS: modified Rodnan Skin Score; DAI: Disease Activity Index; DSS: Disease Severity Scale; ESR: Erythrocyte Sedimentation Rate; CRP: C-Reactive Protein; sPAP: systolic Pulmonary Arterial Pressure; DLco: Diffusing Capacity of the lung for monoxide carbon. *: statistically significant.

## Data Availability

The data presented in this study are available upon reasonable re-quest to the corresponding author.

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
