# Peer review of "Increased Complement Activation in Systemic Sclerosis Patients with Skin and Lung Fibrosis"

_jpm, 2022, doi:10.3390/jpm12020284_

Round 1
Reviewer 1 Report
The authors present interesting work. Main concern is the proper usage of English words and grammar.
Here few examples
Line 26. The influence of complement system in the phenotypic expression of systemic sclerosis (SSc) is a debated topic. We aimed to evaluate complementary fractions in SSc patients and to correlate them with manifestations of disease.
Rewrite the sentence. The use of complementary here and in the text is confusing.
Line 79. At the same time, the activation of immune system conducts 80 to the releasing of proinflammatory cytokine that provokes vascular injury and consecutive fibroblast activation and 81 proliferation with deposition of collagen [3-6].
Rewrite the sentence.
Line 91 . Moreover, an alteration of Toll like receptor (TLR) signaling in SSc patients has been 92 found, suggesting that TLR pathways, and consecutive complement activation and hypocomplementaemia, may play a 93 role in the pathogenesis of the disease. However is still unclear if complement activation and consumption may correlate 94 with disease activity [13]. Rewrite the sentence. Hypocomplementaemia?
Line 250. Table 3. Linear correlation (correlation coefficients=r) between complementary fractions and demographic and disease 251 variables.
Complementary?
Line 300. In this study we also show that total complementary activity is associated with ILD, assessed by both DLco and HRCT. At our knowledge there are not data in literature about complement in the ILD.
Rewrite the sentence. Complementary?
Author Response
We gratefully thank the Reviewer for the suggestion and we deeply revised English text. We corrected/modified all the sentences pointed out by the Reviewer.
Reviewer 2 Report
This is an interesting study assessing the total complement activity in SSc patients. Albeit preliminary, these data open new perspectives in understanding SSc pathogenesis. The manuscript deserves further clarification:
- In Figure 1 A please provide individual values of HC, dcSSc and lcSSc
- regarding the association with ILD, was it independent association observed in multivariate analysis ?
- Were the patients treated ? Please adjust the data on treatments in particular on steroids which could change complement activity
- Could you test the association with PAH, the remaining association with DLCO and not FVC suggests the implication of PAH
- Although the correlations are low there is association with disease activity as reflected by inflammatory markers, is there any correlation with joint or tendon involvement, which are also observed in active disease ?
- The association of CH50 with disease severity assessed by the DSS remains in multivariate analysis:
- i suggest also to test association of CH50 with markers of severe lung fibrosis: association with FVC, with extent of ILD as well as with extensive disease according to Goh
- I suggest to assess independently each of the component of the DSS to determine whether the association is driven by a specific organ involvement
- To better delineate the interest of the complement as a biomarker of severity, i suggest building multivariate models including age, sex, disease duration, antibody status and cutaneous form to measure the weight of a high complement activity in DSS
- To increase the robustness of the biomarker i suggest to use prospective data if available to assess whether a high complement activity is predictive of disease progression
- In discussion, the sentence "we can hypothesize that total complement activity may have a role" is too strong as compared to the findings and the low correlations observed, please reformulate and discuss the low correlation coefficients
- References 1-3 are missing
Author Response
# This is an interesting study assessing the total complement activity in SSc patients. Albeit preliminary, these data open new perspectives in understanding SSc pathogenesis. The manuscript deserves further clarification:
In Figure 1 A please provide individual values of HC, dcSSc and lcSSc
Authors reply: We gratefully thank the Reviewer for the salutary comments that allow us to ameliorate the value of our paper. We modified Figure 1A according to her/his suggestion.
# Regarding the association with ILD, was it independent association observed in multivariate analysis?
Authors reply: We inserted in multiple linear regression analysis DLco as parameter of ILD (see table 4).
# Were the patients treated ? Please adjust the data on treatments in particular on steroids which could change complement activity
Authors reply: We thank the Reviewer for the suggestion to improve our manuscript. All patients enrolled in this study underwent vasodilator therapy with endoprost and none of them underwent immunosuppressive therapy or high dose corticosteroids (equivalent to prednisone>10 mg) in the past 6 months before the enrollment. All patients continued stable therapy from enrollment until clinical evaluation and serum complement assay. Corticosteroids were used by 30 SSc patients (5 mg of prednisone daily). According to Reviewer suggestion we performed Mann-Whitney's test to analyze the association with steroid therapy and we did not find any difference in CH50, C2, C3 and C4 serum level according to steroids. We clarified the therapy regimen in material and method section and added these results in the result section.
# Could you test the association with PAH, the remaining association with DLCO and not FVC suggests the implication of PAH
Authors reply: We thank the Reviewer for the suggestion to improve our manuscript. We performed Mann-Whitney's test to analyze the association with PAH and we did not find any difference in CH50, C2, C3 and C4 serum level according to PAH. We added these results in the result section.
# Although the correlations are low there is association with disease activity as reflected by inflammatory markers, is there any correlation with joint or tendon involvement, which are also observed in active disease?
Authors reply: We thank the Reviewer for the suggestion to improve our manuscript. We performed Mann-Whitney's test to analyse the association with joint/TFR and we did not find any difference in CH50, C2, C3 and C4 serum level according to joint/TFR. We added these results in the result section.
# The association of CH50 with disease severity assessed by the DSS remains in multivariate analysis:
I suggest also to test association of CH50 with markers of severe lung fibrosis: association with FVC, with extent of ILD as well as with extensive disease according to Goh
Authors reply: We thank the Reviewer for the suggestion to improve our manuscript. We did not find any correlation between FVC and CH50. We added this result in the result section and in table 2. Unfortunately we didn’t used computational platform for texture analysis of ILD patterns to quantify parenchymal lung abnormalities and we recognize it as a limitation of the study. We added this sentence in the discussion section.
# I suggest to assess independently each of the component of the DSS to determine whether the association is driven by a specific organ involvement
Authors reply: We thank the Reviewer for the suggestion to improve our manuscript. We assessed independently each of the component of the DSS (general state, peripheral vessels, skin, joints/tendons, muscles, gastrointestinal tract, lungs, heart and kidneys) and CH50 serum levels were significantly associated only with the items skin and lung. We have already analyzed mRSS for skin involvement and DLco/ILD for lung involvement (see result section).
# To better delineate the interest of the complement as a biomarker of severity, i suggest building multivariate models including age, sex, disease duration, antibody status and cutaneous form to measure the weight of a high complement activity in DSS
Authors reply: We thank the Reviewer for the suggestion to improve our manuscript. We performed a multivariate analysis including these variables and we did not find any significant value.
# To increase the robustness of the biomarker I suggest to use prospective data if available to assess whether a high complement activity is predictive of disease progression
Authors reply: We thank the Reviewer for the suggestion. Unfortunately, no prospective data are yet available to assess the predictive value of complement activity in disease progression. This might be an interesting suggestion to confirm the strengthen of our results.
# In discussion, the sentence "we can hypothesize that total complement activity may have a role" is too strong as compared to the findings and the low correlations observed, please reformulate and discuss the low correlation coefficients
Authors reply: According to the Reviewer we ameliorate, the sentence e clarify the low correlation coefficients.
References 1-3 are missing
Authors reply: We added the missing references.
Round 2
Reviewer 1 Report
The authors have revised the manuscript as requested
Reviewer 2 Report
All my comments have been taken into account. I don't have any comment. In my opinion, the manuscript can be accepted in its present form.